# DO DEEP REINFORCEMENT LEARNING ALGORITHMS REALLY LEARN TO NAVIGATE?

## ABSTRACT

Deep reinforcement learning (DRL) algorithms have demonstrated progress in learning to find a goal in challenging environments. As the title of the paper by Mirowski et al. (2016) suggests, one might assume that DRL-based algorithms are able to "learn to navigate" and are thus ready to replace classical mapping and path-planning algorithms, at least in simulated environments. Yet, from experiments and analysis in this earlier work, it is not clear what strategies are used by these algorithms in navigating the mazes and finding the goal. In this paper, we pose and study this underlying question: are DRL algorithms doing some form of mapping and/or path-planning? Our experiments show that the algorithms are not memorizing the maps of mazes at the testing stage but, rather, at the training stage. Hence, the DRL algorithms fall short of qualifying as mapping or path-planning algorithms with any reasonable definition of mapping. We extend the experiments in Mirowski et al. (2016) by separating the set of training and testing maps and by a more ablative coverage of the space of experiments. Our systematic experiments show that the NavA3C+$D_1D_2L$ algorithm, when trained and tested on the same maps, is able to choose the shorter paths to the goal. However, when tested on unseen maps the algorithm utilizes a wall-following strategy to find the goal without doing any mapping or path planning.

## 1 INTRODUCTION

Navigation remains a fundamental problem in mobile robotics and artificial intelligence (Smith & Cheeseman (1986); Elfes (1989)). The problem is classically addressed by separating the task of navigation into two steps, exploration and exploitation. In the exploration stage, the environment is represented as some kind of *map*. In the exploitation stage, the map is used to *plan a path* to a given destination based on some optimality criterion. This classical approach has been quite successful in navigation using a variety of sensors. However, navigation in general unstructured environments, especially with texture-less Yang et al. (2016), transparent and reflective surfaces Lai et al. (2011), remains a challenge.

Recently, end-to-end navigation methods—which attempt to solve the navigation problem without breaking it down into separate parts of mapping and path-planning—have gained traction. With the recent advances in Deep Reinforcement Learning (DRL), these end-to-end navigation methods, such as Mnih et al. (2016); Silver et al. (2016); Levine et al. (2017); Mirowski et al. (2016); Oh et al. (2016), forego decisions about the details that are required in the intermediate step of mapping. The potential for simpler yet more capable methods is rich; for example, the resulting trained agents can potentially optimize the amount of map information required for navigation tasks. One such algorithm by Mirowski et al. (2016) has shown promise in exploring and finding the goal efficiently within complex environments. Notably, this is done using only monocular first-person views.

Despite such potential advances, DRL-based navigation remains a relatively unexplored field with its own limitations. The black-box nature of these methods make them difficult to study, and the patterns captured by the methods are not well understood. Recent work analyzing neural networks has shown that deep learning-based object detection methods can be easily fooled by introducing noise that is imperceptible to humans (Nguyen et al. (2015)); this level of sensitivity motivates why it is particularly important to analyze DRL methods across a wide variety of experiments: we need to understand their strengths and limitations.

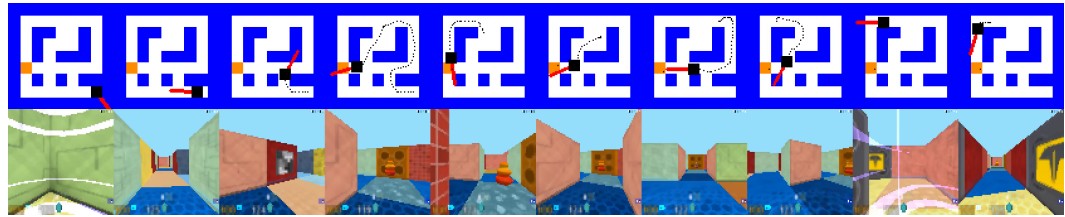

Figure 1: Snapshots of the path taken by the agent while evaluating the model trained on the same random map with random goal and random spawn. The first row shows the top view of the robot moving through the maze with the goal location marked orange, the agent marked black and the agent's orientation marked red. The second row shows the first person view, which, besides reward, is the only input available to the agent and the top view is available only for human analysis.

In this work, we develop a better understanding of recent DRL-based methods. In particular, we thoroughly explore and analyze the state-of-the-art Mirowski et al. (2016) methods across hundreds of maps with increasing difficulty levels. We set up the environment as a randomly generated map, as shown in Fig 1, with an agent and a goal. The agent is provided only with the first-person view and is tasked to find the goal as many times as possible within a fixed amount of time, re-spawning its location each time it reaches the goal. We train and evaluate the algorithms with increasing difficulty. In the easiest stage, we keep the goal location, spawn location and map constant over the training and testing. We call this set up *static goal, static spawn, and static map*. To increase the difficulty, we incrementally randomize the spawn locations, goal locations and map structures until all three are random. We discuss the design of experiments in Section 4.1 in more detail.

Mirowski et al. (2016) do train and test their algorithms with randomized goals and spawns and show that their algorithm is able to exploit the knowledge of the goal location at evaluation time to maximize reward. However, following training and testing on constant map structures, this state-of-the-art result is shown to be successful on only one map, which brings into question the repeatability of the results. It is also unclear whether these results generalize to unseen maps.

Although disjoint training and testing sets are standard practice in machine learning, to the best of our knowledge, we are the first to evaluate any DRL-based navigation method on maps with unseen structures. We expand on the analysis in Mirowski et al. (2016) to address its limitations and ask whether DRL-based algorithms such as NavA3C+$D_1D_2L$ perform any mapping followed by shortest path planning. Our experiments show no evidence of mapping in cases where algorithms are evaluated on unseen maps and no evidence of optimal path planning, even when the map is constant and only the goal is randomized.

To better understand navigation, we compute attention-maps for models to show which portions of the input image are being used. We find that the models discard most of the image information, focusing attention on a small band in the middle of the image except around junctions, in which case the attention is distributed evenly throughout the image.

These findings result from training and testing on multiple maps that were randomly selected from a set of 1100 randomly generated maps. We provide experimental results on ten randomly selected maps and a testing set of 100 unseen maps to ensure results are independent of map choice. We will make our code and data available following the blind review process.

## 2 RELATED WORK

**Localization and mapping**  Localization and mapping for navigation is a classic problem in mobile robotics and sensing. Smith & Cheeseman (1986) introduced the idea of propagating spatial uncertainty for robot localization while mapping, and Elfes (1989) popularized Occupancy Grids. In the three decades since these seminal works, the field has exploded with hundreds of algorithms for many types of sensors (e.g., cameras, laser scanners, sonars and depth sensors). These algorithms vary by how much detail is captured in their respective maps. For example, topological maps, like Kuipers (1978), aim to capture as little information as possible while occupancy grid maps, Elfes (1989), aim to capture metrically accurate maps in resolutions dependent upon the navigation task.

All these approaches require significant hand-tuning for the environment, sensor types and navigation constraints of the hardware. In contrast, end-to-end navigation algorithms optimize the detail of map storage based on the navigation task at hand, which makes them worth exploring.

**Deep reinforcement learning**   DRL gained prominence recently when used by Mnih et al. (2013; 2015) to train agents that outperform humans on Atari games; agents that trained using only the games' visual output. More recently, DRL has been applied to end-to-end navigation (Oh et al. (2016); Mirowski et al. (2016); Chaplot et al. (2016)). It is common for agents to be trained and tested on the same maps with the only variation being the agent's initial spawn point and the map's goal location (Mirowski et al. (2016); Zhu et al. (2017); Kulkarni et al. (2016)).

In contrast, Oh et al. (2016) test their algorithm on random unseen maps, but their agents are trained to choose between multiple potential goal locations based on past observations. The episodes end when the agent collects the goal, so there is no requirement for the algorithm to store map information during their exploration. Thus, their agents decide to avoid a goal of a particular color while seeking other colors rather than remembering the path to the goal. Chaplot et al. (2016) test their method on unseen maps in the VizDoom environment, but only vary the maps with unseen textures. Thus, their agents are texture invariant, but train and test on maps with the same geometric structure.

In this work, we extend the study of these methods in a more comprehensive set of experiments to address the question of whether DRL-based agents remember enough information to obviate mapping algorithms or may in fact need to be augmented with mapping for further progress.

## 3   BACKGROUND

Our problem formulation is based on the work of Mirowski et al. (2016). For completeness, we summarize the technical setup here. For additional details regarding the setup, we refer the reader to Mnih et al. (2016); Mirowski et al. (2016).

The problem of navigation is formulated as an interaction between an environment and an agent. At time $t$ the agent takes an action $a_t \in \mathcal{A}$ and observes observation $o_t \in \mathcal{O}$ along with a reward $r_t \in \mathbb{R}$. We assume the environment to be a Partially Observable Markov Decision Process (POMDP). In a POMDP, the future state of the environment, $s_{t+1} \in \mathcal{S}$, is conditionally independent of all the past states, $s_{1:t-1}$, given the current state $s_t$. It is further assumed that $o_t$ and $r_t$ are independent of previous states given current state $s_t$ and last action $a_{t-1}$. Formally, a POMDP is defined as a six tuple $(\mathcal{O}, C, \mathcal{S}, \mathcal{A}, T, R)$ that is composed of an observation space $\mathcal{O}$, an observation function $C(s_t, a_t) \rightarrow o_t$, a state space $\mathcal{S}$, an action space $\mathcal{A}$, a transition function $T(s_t, a_t) \rightarrow s_{t+1}$ and a reward function $R(s_t, a_t) \rightarrow r_{t+1}$. For our problem setup, the observation space $\mathcal{O}$ is the space of an encoded feature vector that is generated from input image along with previous action and reward. Action space $\mathcal{A}$ contains four actions: rotate left, rotate right, move forward and move backward and reward function $R$ is defined for each experiment so that reaching the goal location leads to high reward with auxiliary rewards to encourage certain kinds of behavior.

For DRL algorithms, the state space $\mathcal{S}$ is not hand tuned, but it is modeled as a vector of floats. Additionally, instead of modeling observation function $C(s_t, a_t) \rightarrow o_t$ and $T(s_t, a_t) \rightarrow s_{t+1}$, a combined transition function $T_c(s_t, o_t, a_t, r_t; \theta_T) \rightarrow s_{t+1}$ is modeled to estimate the next state $s_{t+1}$ and directly take previous observation and reward into account. For policy-based DRL a policy function $\pi(a_{t+1}|s_t, o_t, a_t, r_t; \theta_\pi) \rightarrow \pi_t(a_{t+1}; \theta_\pi)$ and a value function $V(s_t, o_t, a_t, r_t; \theta_V) \rightarrow V_t(\theta_V)$ are also modeled. All three functions $T_c$, $\pi_t$ and $V_t$ share most of the parameters such that $\theta_T \subseteq \theta_\pi \cap \theta_V$

The DRL objective is to estimate unknown weights $\theta = \theta_T \cup \theta_\pi \cup \theta_V$ that maximizes the expected future reward $R_t = \sum_{k=t}^{t_{end}-t} \gamma^{k-t} r_k$ (where $\gamma$ is the discount factor) and is expressed as

$$\theta^* = \arg\max_\theta \mathbb{E}[R_t], \tag{1}$$

where $\mathbb{E}[.]$ denotes the expected value.

**Asynchronous Advantage Actor-Critic**   In this work, we use the policy-based method called Asynchronous Advantage Actor-Critic (A3C) (Mnih et al. (2016)), which allows weight updates to happen asynchronously in a multi-threaded environment. It works by keeping a "shared and slowly changing copy of target network" that is updated every few iterations by accumulated gradients in

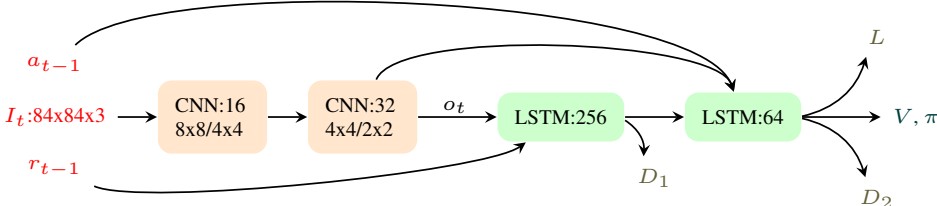

Figure 2: Modified NavA3C+$D_1D_2$L (Mirowski et al. (2016)) architecture. The architecture is has three inputs the current image $I_t$ and previous action $a_{t-1}$ and previous reward $r_{t-1}$. As shown by Mirowski et al. (2016), the architecture improves upon vanilla A3C architecture by using auxiliary outputs of loop-closure signal $L$ and predicted depth $D_1$ and $D_2$. Since we use a smaller action space than Mirowski et al. (2016) and our agent moves with constant velocity, we do not use velocity at previous time step as input signal.

each of the threads. The gradients are never applied to the local copy of the weights; instead, a local copy of weights is periodically synced from the shared copy of target weights. The gradient for the weight update is proportional to the product of *advantage*, $R_t - V_t(\theta_V)$, and *characteristic eligibility*, $\nabla_{\theta_\pi} \ln \pi_t(a_{t+1}; \theta_\pi)$ (Williams (1992)), which update the weights as

$$\theta_\pi \leftarrow \theta_\pi + \sum_{t \in \text{episode}} \alpha_\pi \nabla_{\theta_\pi} \ln \pi_t (R_t - V_t(\theta_V)) \tag{2}$$

$$\theta_V \leftarrow \theta_V + \sum_{t \in \text{episode}} \alpha_V \frac{\partial (R_t - V_t(\theta_V))^2}{\partial \theta_V} . \tag{3}$$

For additional details of the A3C algorithm, we refer the reader to Mnih et al. (2016).

**NavA3C+$D_1D_2$L** In this work, we use the NavA3C+$D_1D_2$L architecture as proposed by Mirowski et al. (2016), which builds modifying the network architecture to have two LSTMs and with auxiliary outputs of depth predictions along with loop-closure predictions. The schematic of the architecture is shown in Fig 2. The architecture has three inputs: the current image $I_t$, previous action $a_{t-1}$ and previous reward $r_{t-1}$. As shown by Mirowski et al. (2016), the architecture improves upon vanilla A3C architecture by optimizing predictions for the auxiliary outputs of loop closure signal $L$ and predicted depth $D_1$ and $D_2$. Since we use a smaller action space than Mirowski et al. (2016) and our agent moves with constant velocity, we do not use velocity at the previous time step as an input signal.

## 4 THE DRL NAVIGATION CHALLENGE

Since deep reinforcement learning algorithms need millions of iterations to train, in the absence of thousands of robotic replicas like Levine et al. (2017), we evaluate the algorithms on a simulated environment. We use the same game engine as Mirowski et al. (2016), called Deepmind Lab (Beattie et al. (2016)). The game is setup such that an agent is placed within a randomly generated maze containing a *goal* at a particular location. On reaching the goal, the agent *re-spawns* within the same maze while the goal location remains unchanged. Following Mirowski et al. (2016), we scatter the maze with randomly placed smaller apple rewards (+1) to encourage initial explorations and assign the goal a reward of +10. The agent is tasked to find the goal as many times as possible within a fixed amount of time, re-spawning within the maze, either statically or randomly, each time it reaches the goal.

Unlike Mirowski et al. (2016), we include a small wall penalty (-0.2) that pushes the agent away from the wall. The wall penalty is useful to prevent agents from moving along the walls, thereby discarding vision input for exploration. We also use a discrete 4-action space (move forward/backward, rotate left/right)which is different from the 8-action space one used by Mirowski et al. (2016). A smaller action space helps the algorithm train faster while achieving similar reward values.

| Map ID 127 | Map ID 169 | Map ID 246 | Map ID 336 | Map ID 445 | Map ID 589 | Map ID 691 | Map ID 828 | Map ID 844 | Map ID 956 |

Figure 3: The ten randomly chosen mazes for evaluation. We generate 1100 random mazes and choose ten to evaluate our experiments that require testing and training on the same maps.

We generate 1100 random maps using depth-first search based maze generation methods. More information on maze generation can be found in the appendix. Of the first 1000 maps, 10 are randomly selected for our static-map experiments (Fig. 3). For our unseen map experiments, agents are trained on increasing subsets of the first 1000 maps and tested on the remaining 100. Unlike Mirowski et al. (2016) and similar to Chaplot et al. (2016), we use randomly textured walls in our mazes so that the policies learned are texture-independent.

## 4.1 EXPERIMENTS

We evaluate the NavA3C+$D_1D_2L$ algorithm on maps with 5 stages of difficulty. While the algorithm works smoothly on the easier stages, it does not perform better than wall-following methods on the hardest stage. We propose these experiments as a 5-stage benchmark for all end-to-end navigation algorithms.

1. **Static goal, static spawn, and static map** To perform optimally on this experiment, the agent needs to find and learn the shortest path at training time and repeat it during testing.

2. **Static goal, random spawn and static map** This is a textbook version of the reinforcement learning problem, especially in grid-world Sutton & Barto (1998), with the only difference being that the environment is partially observable instead of fully observable. This problem is more difficult than Problem 1 because the agent must find an optimal policy to the goal from each possible starting point in the maze.

3. **Random goal, static spawn, and static map** In this setup, we keep the spawn location and the map fixed during both training and testing but choose a random goal location for each episode. Note that the goal location stays constant throughout an episode. The agent can perform well on this experiment by remembering the goal location after it has been discovered and exploiting the information to revisit the goal faster.

4. **Random goal, random spawn, and static map** In this version of the experiment both the spawn point and the goal location is randomized. To perform optimally, the agent must localize itself within the map in addition to being able to exploit map-information.

   This is the problem that is addressed by Mirowski et al. (2016) with limited success. They evaluate this case on two maps and report Latency $1 :> 1$ to be greater than 1 in one of the two maps. We evaluate the same metric on ten other maps.

5. **Random goal, random spawn, and random map** We believe that any proposed algorithms on end-to-end navigation problems, should be evaluated on unseen maps. To our knowledge, this is the first paper to do so in the case of deep reinforcement learning based navigation methods. We train agents to simultaneously learn to explore 1, 10, 100, 500 and 1000 maps and test them on the same 100 unseen maps. The relevant results can be found in Fig 5 and discussed in Section 5.

The comparative evaluation of the different the stages of this benchmark are shown in Fig 4 and expanded upon in the Section 5.

## 4.2 EVALUATION METRICS

We evaluate the algorithms in terms of three metrics: rewards, *Latency* $1 :> 1$ and *Distance-inefficiency*. Following Mirowski et al. (2016), we report *Latency* $1 :> 1$, a ratio of the time taken to hit the goal for the first time (exploration time) versus the average amount of time taken to hit goal subsequently (exploitation time). The metric is a measure of how efficiently the agent exploits map information to find a shorter path once the goal location is known. If this ratio is greater than

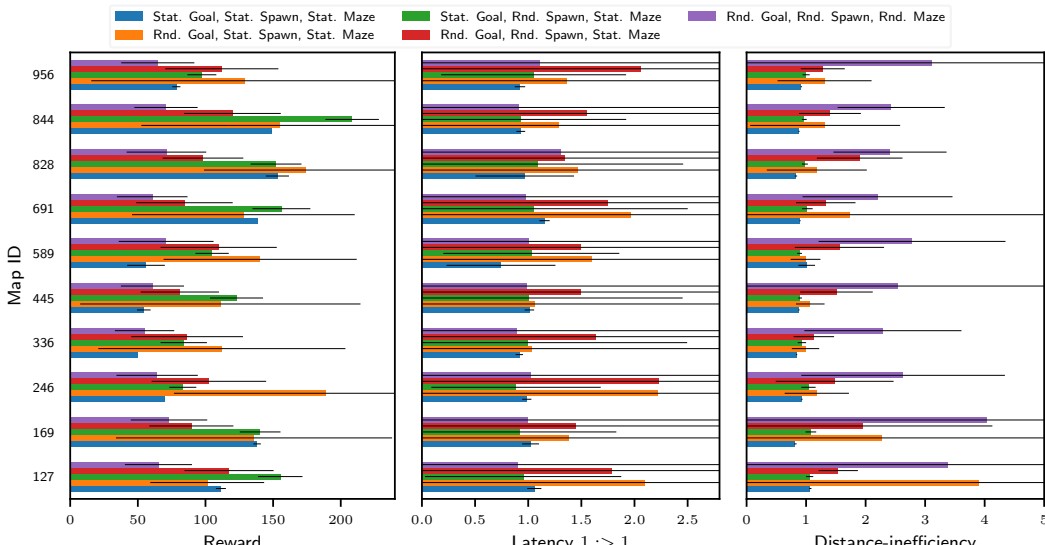

Figure 4: We evaluate the NavA3C+$D_1D_2$L Mirowski et al. (2016) algorithm on ten randomly chosen maps, shown in Fig. **??**, with increasing difficulty as described in Sec. 4.1. The figure is best viewed in color. Vertical axis is one of the ten map ID's on which the agent was trained (except for Rnd. Maze) and evaluated. Horizontal axis are different evaluation metrics. We note that when the goal is static then rewards are consistently higher as compared to random goal. With static goals, the metric Distance-inefficiency is close to 1, indicating that the algorithms are able to find shortest path. However, with random goals, the agents struggle to find the shortest path. From the Latency $1 :> 1$ results we note that the algorithm do well when trained and tested on the same map but fail to generalize to new maps when evaluated on ability to exploit the information about goal location. Also note that Latency $1 :> 1$ metric for cases of static goals is expected to be close to one because the location of goal is learned at train time.

1, the agent is doing better than random exploration and the higher the value, the better its map-exploitation ability. Note that the metric is meaningful only when the goal location is unknown at evaluation time.

*Distance-inefficiency* is defined to be the ratio of the total distance traveled by the agent versus the sum of approximate shortest distances to the goal from each spawn point. The metric also disregards goals found during exploitation time as the agent must first find the goal before it can traverse the shortest path to it. Note that the shortest distance between the spawn and goal locations is computed in the top-down block world perspective and hence is only an approximation.

While the Latency $1 :> 1$ measures the factor by which planned path to the goal is shorter than the exploration path, the Distance-inefficiency measures the length of this path with respect to the shortest possible path.

## 5 RESULTS AND ANALYSIS

In this section we discuss the results for experiments as discussed in Section 4.1 over ten randomly chosen maps shown in Fig 3. The results in Fig 4.

**Static goal, static spawn, static maze** For this case, the reward is consistently high, and Distance-inefficiency is close to 1 with small standard deviations implying the path chosen is the shortest available. Please note that Latency $1 :> 1$, is should be close to 1 for static goal case, because the goal location is known at training time.

**Static goal, random spawn, static map** Again, note that Distance-inefficiency is close to 1 implying that when the goal is found, the shortest path is traversed. This is because the agent can learn the optimal policy for shortest path to the goal at train time.

**Random goal, static spawn, static map**  In this case, the mean of the Latency $1 :> 1$ is more than 1 show that in general the agent is able to exploit map information. However the large standard deviations in this metric and the reward values show that this exploitation is not consistent through episodes. For most of the experiments, the Distance-inefficiency is close to one within error metrics, again imply that the shortest path is taken when the goal is found.

**Random goal, Random spawn, static map**  Similar to the previous experiment, the Latency $1 :> 1$ is more than 1 but with a large standard deviation implying inconsistent performance from episode to episode. The Distance-inefficiency is larger than 1 showcasing the paths traversed to the goal are not necessarily the shortest.

**Random goal, Random spawn, Random map**  For this experiment, agents trained on a 1000 maps are tested individually on the 10 chosen maps that are a subset of the 1000 maps. The Latency $1 :> 1$ is close to 1 implying that map-exploitation is taking place. The large Distance-inefficiency numbers seem to confirm this statement. We present, qualitative results in Sec. 5.3 on very simple, to show that the agents are only randomly exploring the maze rather than utilizing shortest path planning.

## 5.1 EVALUATION ON UNSEEN MAPS

The results for training on $N$ maps, where $N \in \{10, 100, 500, 1000\}$, and testing on 100 unseen maps are shown in Fig 5. We observe that there is a significant jump of average reward and average goal hits when the number of training maps is increased from 10 to 100 but no significant increase when the number of training maps are increased from 100 to 500 to 1000. This is due to the fact that the wall-following strategy learned by the algorithm, is learned with enough variation in 100 maps and training on additional maps does not add to the learned strategy.

## 5.2 EFFECT OF APPLES AND TEXTURE

We evaluate the effect of apples and texture during evaluation time in Fig 5. We train the algorithm on randomly chosen training maps with random texture and evaluate them no maps with and without random texture and also on maps with and without apples. When the apples are present, we place the apples with probability 0.25 in each block of the map. We find that the algorithm, being trained on random textures and random placement of apples, is robust to presence or absence of textures and apples.

## 5.3 QUALITATIVE EVALUATION ON SIMPLE MAPS

To evaluate what strategies that the algorithm is employing to reach the goal we evaluate the algorithm on very simple maps where there are only two paths to reach the goal. The qualitative results for the evaluation are shown in Fig 6.

**Square map**  A Square map (Fig 6) is the simplest possible map with two paths to the goal. We evaluate the algorithm trained on 1000 random maps on square map. We observe that the agent greedily moves in the direction of initialization. This may be because of the initial learning which is motivated by small rewards of getting apples. We compute the percentage of times the agent takes the shortest path over a trial of 100 episodes. We find the agent takes the shortest path only 50.4% ($\pm 12.8\%$) of the times, no better than random.

**Wrench map**  To eliminate the dependency on initial orientation, we evaluate the algorithm on Wrench map as shown in Fig 6. We fix in the spawn point at the bottom of the tail so that shortest path is independent of the spawn orientation. The decision about the shortest path is made at the junction where the agent can either chose to go left or right. We find that the agent is taking shortest path only 32.9% ($\pm 25.1\%$) of the times which is again to better than random.

**Goal map**  Similarly to the wrench map, the goal map (Fig 6) provides a decision point independent of the initial orientation, but it penalizes the wrong decision more than the wrench map 42.6% ($\pm 35.1\%$) of the times which is again no better than random.

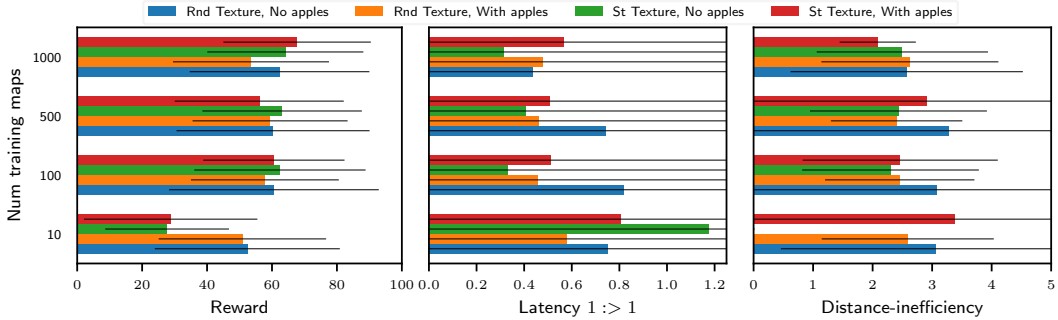

Figure 5: Plots showing the effect of number of training maps with random texture (Rnd Texture) and presence of apples (With apples), when evaluated on unseen maps. We note that although the difference between mean metrics is negligible as compared to standard deviation of the metrics. Hence we say that the effect of apples or textures can be ignored. The only clear trend is apparent Latency $1 :> 1$ metric which suggest that random texture along without apples is advantageous in exploiting goal location while finding the goal second time on-wards.

These experiments show that NavA3C+$D_1D_2L$ algorithm, even when trained on 1000 maps, do not generalize to these very simple maps. Again note that even in cases when there are only two possible paths to the goal, the agent is unable to chose the shorter path with more than 50% probability. This shows that the models trained on 1000 maps have learned only a wall-following strategy rather than learning to plan path based on goal location.

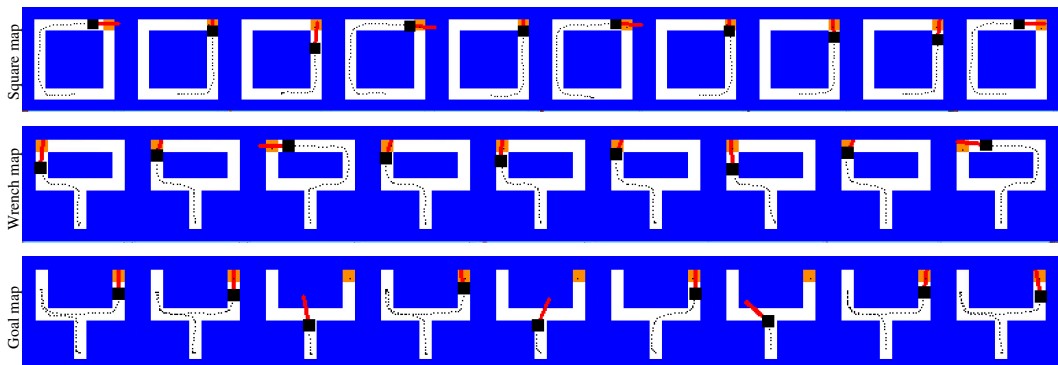

Figure 6: Snapshots of path taken by the agent to reach the goal in a single episode when model trained on 1000 maps is evaluated Square, Wrench and Goal map. The top row shows an evaluation example on Square map, the agent takes the shortest path 6/10 times but when averaged over 100 episodes, the percentage of shortest path taken is not better than random 50.4% ($\pm$12.8%). Although for the example of Wrench map the agent takes the shortest path 8/10 times but when averaged over 100 episodes, the percentage of shortest path taken is reduced to 32.9% ($\pm$25.1%). For the Goal map, the example chosen here shows that the shortest path is only taken 1/6 times, on an average over 100 episodes, the shortest path is taken 42.6% ($\pm$35.1%) times.

## 5.4 ATTENTION HEAT MAPS

We use the normalized sum of absolute gradient of the loss with respect to the input image as a proxy for attention in the image. The gradients are normalized for each image so that the maximum gradient is one. The attention values are then used as a soft mask on the image to create the visualization as shown in Fig 7

We observe that the attention is uniformly distributed on the image when the agent spawns. The attention narrows down to a few pixels in the center when the agent is navigating through the corridor. It spreads to the entire image around turns and junctions. The attention also pays close attention to important objects like goal, apples and unique decals.

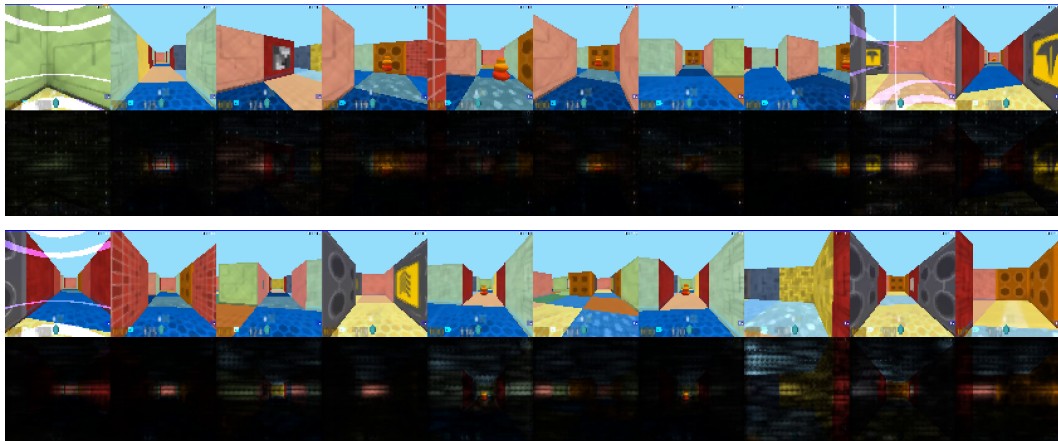

Figure 7: Visualizing attention for two sequences. The first two rows show the sequence when the model is trained on and evaluated on the same map. The last two rows shows the sequence for a model trained on 1000 maps and evaluated on one of the maps. We observe that the attention is uniformly distributed on the image when the agent spawns. The attention narrows down few pixels in the center when the agent is navigating through the corridor. It spreads to the entire image around turns and junctions. The algorithm also pays close attention to important objects like goal, apples and unique decals.

## 6 CONCLUSION

In this work, we comprehensively evaluate NavA3C+$D_1D_2L$ (Mirowski et al. (2016)), a DRL-based navigation algorithms, through systematic set of experiments that are repeated over multiple randomly chosen maps. Our experiments show that DRL-based navigation models are able to perform some degree of path-planning and mapping when trained and tested on the same map even when spawn locations and goal locations are randomized. However the large variation in the evaluation metrics show that how such behaviour is not consistent across episodes. We also train and test these methods on disjoint set of maps and show that such trained models fail to perform any form of path-planning or mapping in unseen environments.

In this work, we begin by asking: do DRL-based navigation algorithms really "learn to navigate"? Our results answer this question negatively. At best, we can say that DRL-based algorithms learn to navigate in the exact same environment, rather than general technique of navigation which is what classical mapping and path planning provide. We hope that the systematic approach to the experiments in this work serve as a benchmark for future DRL-based navigation methods.

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

APPENDIX

HYPERPARAMETERS

We use the Deepmind Lab environment to train our experiments. As mentioned previously, apple rewards are scattered throughout the maze and constitue a +1 reward. Goals constitute a +10 reward. An included wall penalty linearly penalizes the agent as it moves closer to the wall with the penalty being capped off at -.2 per frame. Our episodes are of fixed time length ending at 40 seconds each. The agent interacts with the environment at a rate of 30 frames per second. Each episode thus consists of 1200 frames of data coupled with the corresponding reward signals. Our mazes constitute an area of $900units \times 900units$ though we provide the tools to generate mazes to arbitrary dimensions.

Our A3C implementation is a modified version of OpenAIs open-sourced *universe-starter-agent*. RGB images are fed in to the network of dimensions $84 \times 84 \times 3$. 16 threaded agents are used for all experiments. We use a learning rate of $10^{-4}$ along with the *AdamOptimizer* to train our network. Our models train for a maximum of $10^8$ iterations though we end them early if maximum reward saturates.

BENCHMARKING CODE

To motivate more comprehensive experimental evaluations of DRL-based navigation methods, we will be releasing all our trained models coupled with corresponding reward curves and videos of performance online. This will include completely reproducible evaluation sets wherein we display metric scores for all the trained models on the follow environments:

- the original training conditions
- the training conditions in the absence of apples and textures
- the 100 unseen testing maps
- the planning maps i.e. the square, wrench and goal map

We hope our work can also be utilized as a stepping stone for the creation of better generalized DRL navigation methods bypassing the needless amounts of time spent engineering the infrastructure necessary for these experiments. All our work will be available on github after the blind-review process is over.

