# OpenReview forum: "Do Deep Reinforcement Learning Algorithms really Learn to Navigate?"
_ICLR.cc/2018/Conference — Reject_

### Official Review · AnonReviewer1 · 2017-11-27
**Good paper - but could be made even stronger**

**Rating:** 7
**Confidence:** 4

**Review:**

The paper evaluates one proposed Deep RL-based model (Mirowski et al. 2016) on its ability to generally navigate. This evaluation includes training the agent on a set of training mazes and testing it's performance on a set of held-out test mazes. Evaluation metrics include repeated latency to the goal and comparison to the shortest route. Although there are some (minor) differences between the implementation with Mirowski et al. 2016, I believe the conclusions made by the authors are mostly valid.

I would firstly like to point out that measuring generalization is not standard practice in RL. Recent successes in Deep RL--including Atari and AlphaGo all train and test on exactly the same environment (except for random starts in Atari and no two games of Go being the same). Arguably, the goal of RL algorithms is to learn to exploit their environment as quickly as possible in order to attain the highest reward. However, when RL is applied to navigation problems it is tempting to evaluate the agent on unseen maps in order to assess weather the agent has learned a generic mapping & planning policy. In the case of Mirowski et al. this means that the LSTM has somehow learned to do general SLAM in a meta-learning sense. To the best of my knowledge, Mirowski et al. never made such a bold claim (despite the title of their paper).

Secondly, there seems to be a big disconnect between attaining a high score in navigation tasks and perfectly solving them by doing general SLAM & optimal path planning. Clearly if the agent receives the maximal possible reward for a well designed navigation task it must, by definition, be doing perfect SLAM & path planning. However at less than optimal performance the reward fails to quality  the agent's ability to do SLAM. The relationship between reward and ability to do general SLAM is not clear. Therefore it is my opinion that reinforcement learning approaches to SLAM lack a concrete goal in what they are trying to show.

Minor points: Section 5.3 Square map: how much more reward will the agent gain by taking the optimal path? Perhaps not that much? Wrench map: the fact that the paths taken by the agent are not distributed evenly makes me suspicious. Could the authors generate many wrench maps (same topology, random size, random wall textures) to make sure there is no bias?

---

> ### Author Response · Authors · 2017-12-03
> **Thanks for understanding the claims of the paper**
>
> > Minor points: Section 5.3 Square map: how much more reward will the agent gain by taking the optimal path? Perhaps not that much?
>
> OUR RESPONSE: We don't know. Yes, probably not much. We can perform that experiment and that number.
>
> > Wrench map: the fact that the paths taken by the agent are not distributed evenly makes me suspicious. Could the authors generate many wrench maps (same topology, random size, random wall textures) to make sure there is no bias?
>
> OUR RESPONSE: Yes we can add more experiments. However, we do not think that there is anything suspicious about paths being evenly distributed. We think the exploration strategy learned by the agent closely mirrors a randomized version of bug exploration algorithm. We think the bias helps the algorithm avoid taking random turns canceling itself out often.

---

### Official Review · AnonReviewer2 · 2017-11-27
**Logically flawed critique of a specific paper with unsupported broad generalizations**

**Rating:** 3
**Confidence:** 5

**Review:**

Science is about reproducible results and it is very commendable from scientists to hold their peers accountable for their work by verifying their results. It is also necessary to inspect claims that are made by researchers to avoid the community straying in the wrong direction. However, any critique needs to be done properly, by 1) attending to the actual claims that were made in the first place, by 2) reproducing the results in the same way as in the original work, 3) by avoiding introducing false claims based on a misunderstanding of terminology and 4) by extensively researching the literature before trying to affirm that a general method (here, Deep RL) cannot solve certain tasks.

This paper is a critique of deep reinforcement learning methods for learning to navigate in 3D environments, and seems to focus intensively on one specific paper (Mirowski et al, 2016, “Learning to Navigate in Complex Environments”) and one of the architectures (NavA3C+D1D2L) from that paper. It conducts an extensive assessment of the methods in the critiqued paper but does not introduce any alternative method. For this reason, I had to carefully re-read the critiqued paper to be able to assess the validity of the arguments made in this submission and to evaluate its merit from the point of view of the quality of the critique. The (Mirowski et al, 2016) paper shows that a neural network-based agent with LSTM-based memory and auxiliary tasks such as depth map prediction can learn to navigate in fixed environments (3D mazes) with a fixed goal position (what they call “static maze”), and in fixed mazes with changing goal environments (what they call “environments with dynamic elements” or “random goal mazes”).

This submission claims that:
[a] “[based on the critiqued paper] one might assume that DRL-based algorithms are able to 'learn to navigate' and are thus ready to replace classical mapping and path-planning algorithms”,
[b] “following training and testing on constant map structures, when trained and tested on the same maps, [the NavA3C+D1D2L algorithm] is able to choose the shorter paths to the goal”,
[c] “when tested on unseen maps the algorithm utilizes a wall-following strategy to find the goal without doing any mapping or path planning”,
[d] “this state-of-the-art result is shown to be successful on only one map, which brings into question the repeatability of the results”,
[e] “Do DRL-based navigation algorithms really 'learn to navigate'? Our results answer this question negatively.”
[f] “we are the first to evaluate any DRL-based navigation method on maps with unseen structures”

The paper also conducts an extensive analysis of the performance of a different version of the NavA3C+D1D2L algorithm (without velocity inputs, which probably makes learning path integration much more difficult), in the same environments but by introducing unjustified changes (e.g., with constant velocities and a different action space) and with a different reward structure (incorporating a negative reward for wall collisions). While the experimental setup does not match (Mirowski et al, 2016), thereby invalidating claim [d], the experiments are thorough and do show that that architecture does not generalize to unseen mazes. The use of attention heat maps is interesting.

The main problem however is that it seems that this submission completely misrepresents the intent of (Mirowski et al, 2016) by using a straw man argument, and makes a rather unacademic and unsubstantiated accusation of lack of repeatability of the results.

Regarding the former, I could not find any claim that the methods in (Mirowski et al, 2017) learn mapping and path planning in unseen environments, that could support claim [a]. More worryingly, when observing that the method of (Mirowski et al, 2017) may not generalize to unseen environments in claim [c], the authors of this submission seem to confuse navigation, cartography and SLAM, and attribute to that work claims that were never made in the first place, using a straw man argument. Navigation is commonly defined as the goal driven control of an agent, following localization, and is a broad skill that involves the determination of position and direction, with or without a map of the environment (Fox 1998, ” Markov Localization: A Probabilistic Framework for Mobile Robot Localization and Navigation”). This widely accepted definition of navigation does not preclude being limited to known environments only.

Regarding repeatability, the claim [d] is contradicted in section 5 when the authors demonstrate that the NavA3C+D1D2L algorithm does achieve a reduction in latency to goal in 8 out of 10 experiments on random goal, static map and random or static spawns. The experiments in section 5.3 are conducted in simple but previously unseen maps and cannot logically contradict results (Mirowski et al, 2016) achieved by training on static maps such as their “I-maze”. Moreover, claim [d] about repeatability is also invalidated by the fact that the experiments described in the paper use different observations (no velocity inputs), different action space, different reward structure, with no empirical evidence to support these changes. It seems, as the authors also claim in [b], that the work of (Mirowski et al, 2017), which was about navigation in known environments, actually is repeatable.

Additionally, some statements made by the authors are demonstrably untrue. First, the authors claim that they are the first to train DRL agents in all random mazes [f], but this has been already shown in at least two publications (Mnih et al, 2016 and Jaderberg et al, 2016).

Second, the title of the submission, “Do Deep Reinforcement Learning Algorithms Really Learn to Navigate” makes a broad statement [e] that cannot be logically invalidated by only one particular set of experiments on a particular model and environment, particularly since it directly targets one specific paper (out of several recent papers that have addressed navigation) and one specific architecture from that paper, NavA3C+D1D2L (incidentally, not the best-performing one, according to table 1 in that paper). Why did the authors not cite and consider (Parisotto et al, 2017, “Neural Map: Structured Memory for Deep Reinforcement Learning”), which explicitly claims that their method is “capable of generalizing to environments that were not seen during training”? It seems that the authors need to revise both their bibliography and their logical reasoning: one cannot invalidate a broad set of algorithms for a broad goal, simply by taking a specific example and showing that it does not fit a particular interpretation of navigation *in previously unseen environments*.

---

> ### Author Response · Authors · 2017-12-03
> **Criticism based on misunderstanding of the claims of the paper and disagreement on defintion of "navigation"**
>
> > The paper also conducts an extensive analysis of the performance of a different version of the NavA3C+D1D2L .... The use of attention heat maps is interesting.
>
> OUR RESPONSE: The critique here stems from the misunderstanding that we are claiming Mirowski et al. results are not reproducible or false. In fact, we get similar results on static maps with slight different architecture, proving that the work was reproducible. Having said that we do stress that the _Latency 1:>1_ metric result is meaningfully good for only one map in Mirowski et al., not to claim that it is false but to stress the need to evaluate on a bigger set of experiments. Had we not got similar results as Mirowski et al. under similar kind of maps, we would have been pushed to make the architecture exactly same as Mirowski's.
>
> > The main problem however is that it seems that this submission completely misrepresents the intent of (Mirowski et al, 2016) by using a straw man argument, and makes a rather unacademic and unsubstantiated accusation of lack of repeatability of the results.
>
> OUR RESPONSE: We did not make any such claim. At least we did not intend to make any such claim. We never said in our paper that Mirowski et al claimed that their algorithm works on unseen maps.
>
> > Regarding the former, ...  This widely accepted definition of navigation does not preclude being limited to known environments only.
>
> OUR RESPONSE: This part of the criticism seem to arise from disagreement on the definition of the word "navigation". In claim [a] we are careful with our use of words. We believe that there is no agreement on the "widely accepted definition" of the word "navigation." Based on one's understanding of the word "navigation", "one might assume" that the algorithm might generalize to unseen worlds. We think that criticism of our work, based on our choice of definition of "navigation" is unfair. It is even unfair to cite a single paper to impose the reviewers definition of the word.
>
> The other part of the criticism that we use a "straw man" is again wrong because we do not intend to show pathology with Mirowski et al. paper, experiments or claims. In other words we are raising the standards what capability a "learning to navigate" paper should be demonstrating.
>
> > Regarding repeatability, the claim [d] ...  actually is repeatable.
>
> OUR RESPONSE: This criticism is again based on misunderstanding of sentence [d].
> We do not claim that our experiments contradict Mirowski et al. results. We use the sentence [d] as part of our motivation that doing experiments only one random map makes results of scientific work questionable and must be repeated on a larger set of randomized samples. Yes as claimed in [b] our results actually support Mirowski et al.'s results.
>
> > Additionally,....  publications (Mnih et al, 2016 and Jaderberg et al, 2016).
>
> OUR RESPONSE: This criticism again stems from the mis-communication of our claim and disagreement on definition of word "navigation." Our use of the phrase "DRL-based navigation method", implied application of DRL-based method on the task of "navigation". Both Mnih et al. 2016, Jaderberg et al. 2016 evaluate their algorithms on navigation agnostic metrics like cumulative reward or human normalized score instead of navigation specific metrics like "Latency 1:>1" or "distance efficiency". Evaluation on navigation agnostic metric means that agent could be exploring the maze better instead of actually doing better than wall following. Also, Mnih et al, 2016 do generate random maps but they chose a random map and train and test on the same random map. This is different from our claim on testing DRL-based navigation methods on unseen environments which means that test map should be structurally different from train map and the DRL
>
> > Second, .... environments*.
>
> OUR RESPONSE: We accept that our title makes a broad statement just Mirowski et al 2016 did. We did overplay our work which led to all the confusion and misunderstanding. Our claim was not to falsify the results in Mirowski et al 2016. We should have titled the paper "Raising the bar for ``learning to navigate''." We can still do that if reviewers agree. We provided a rigorous set of experiments and metrics that can be used to justify if an algorithm has actually "learned to navigate".
>
> Neural Map is a relevant paper but we feel that is unreasonable to criticize us for not citing and considering an non-peer reviewed ArXiV paper.

---

### Official Review · AnonReviewer3 · 2017-11-28
**Un-conclusive experiments and no proposed improvements over existing methods**

**Rating:** 3
**Confidence:** 4

**Review:**

This paper proposes to re-evaluate some of the methods presented in a previous paper with a somewhat more general evaluation method.

The previous paper (Mirowski et al. 2016) introduced a deep RL agent with auxiliary losses that facilitates learning in navigation environments, where the tasks were to go from a location to another in a first person viewed fixed 3d maze, with the starting and goal locations being either fixed or random. This proposed paper rejects some of the claims that were made in Mirowski et al. 2016, mainly the capacity of the deep RL agent to learn to navigate in such environments.

The proposed refutation is based on the following experiments:
- an agent trained on random maps does much worse on fixed random maps that an agent trained on the same maps its being evaluated on (figure 4)
- when an agent is trained on fixed number of random map, its performance on random unseen maps doesn't increase with the number of training maps beyond ~100 maps. (figure 5). The authors argue that the reason for those diminishing returns is that the agent is actually learning a trivial wall following strategy that doesn't benefit from more maps.
- when evaluated on hand designed small maps, the agent doesn't perform very well (figure 6).

There is addition experimental data reported which I didn't find very conclusive nor relevant to the analysis, particularly the attention heat map and the effect of apples and texture.

I don't think any of the experiments reported actually refute any of the original paper's claim. All of the reported results are what you would expect. It boils down to these simple commonly known facts about deep RL agents:
- When evaluated outside of its training distribution, it might not generalized very well (figure 4/6)
- It has a limited capacity so if the distribution of environments is too large, its performance will plateau (figure 5). By the way to me results presented in figure 5 are not enough to claim that the agent trained on random map is implementing a purely reactive wall-following strategy. In fact, an interesting experiment here would have been to do ablation studies e.g. by replacing the LSTM with a feed forward fully connected network. To me the reported performance plateau with number of map size is normal expected behavior, only symptomatic that this deep RL agent has finite capacity.

I think this paper does not provide compelling pieces of evidence of unexpected pathological behavior in the previous paper, and also does not provide any insight of how to improve upon and address the obvious limitations of previous work. I therefore recommend not to accept this paper in its current form.

---

> ### Author Response · Authors · 2017-12-03
> **Criticism based on miscommunication of the claims of the paper.**
>
> > I don't think any of the experiments reported actually refute any of the original paper's claim. All of the reported results are what you would expect. It boils down to these simple commonly known facts about deep RL agents:
>
> OUR RESPONSE: We do not refute any of the original paper's experimental claims. We question the claim whether the algorithm actually "learns to navigate" as one might mistakenly interpret the title to be. Our contribution is to bring into light the limitations of their algorithm rather than to refute their experimental claims.
>
> > - When evaluated outside of its training distribution, it might not generalized very well (figure 4/6)
> > - It has a limited capacity so if the distribution of environments is too large, its performance will plateau (figure 5). By the way to me results presented in figure 5 are not enough to claim that the agent trained on random map is implementing a purely reactive wall-following strategy. In fact, an interesting experiment here would have been to do ablation studies e.g. by replacing the LSTM with a feed forward fully connected network. To me the reported performance plateau with number of map size is normal expected behavior, only symptomatic that this deep RL agent has finite capacity.
>
> OUR RESPONSE: Yes, those facts are not only true about Reinforcement learning but also about machine learning in general. Those are commonly know facts about machine learning in general. Yet most of the problems in machine learning are about finding models that generalize from training distribution to test distribution. Our experiments show that the NavA3C-D1D2L does not generalize to test distribution. We do not think it is because the model is saturating out. We can definitely add the experiments as the reviewer suggest.
>
> > I think this paper does not provide compelling pieces of evidence of unexpected pathological behavior in the previous paper, and also does not provide any insight of how to improve upon and address the obvious limitations of previous work. I therefore recommend not to accept this paper in its current form.
> > Confidence: 4: The reviewer is confident but not absolutely certain that the evaluation is correct
>
> OUR RESPONSE: We did not claim unexpected pathological behavior in the previous paper. We pointed out the failure cases of the algorithm in what we though a navigation task should be.

---

> > ### Public Comment · (anonymous) · 2017-12-19
> > **Training on random maps**
> >
> > I really find that some of the results in the paper are inspiring. I'd like to provide my thoughts on the possible reason for the results observed when training on random maps.
> >
> > > We do not think it is because the model is saturating out.
> >
> > In contrast with the reviewer's opinion, I also do not think the model is saturating out, either. My understanding is that if the agent is trained on a distribution of random maps, according to the formulation of AC (particularly, the value function is an estimation of the expected future reward), isn't it the case that the agent should learn to perform an "average" of behavior even on a particular test map (especially when the goal is not in the view)? Note that this problem is a POMDP, so the agent can only estimate an average future reward when only the environment is partially observed (imagine in a different map but the agent sees the same thing and the goal is in a different location). Because the training reward is designed in a way that there is always a chance of apple appearing in a grid, then the correct "average" behavior should be wall-following?

---

### Decision · Program_Chairs · 2018-01-29
**ICLR 2018 Conference Acceptance Decision**

**Decision:**

Reject

**Comment:**

This paper received divergent ratings (7, 3, 3). While there is value in thorough evaluation papers, this manuscript has significant presentation issues. As all three reviewers point out, the way it is currently written, it misrepresents the claims made by Mirowski et al 2016 and over-reaches in its findings. Unfortunately, we cannot make a decision on what the manuscript may look like in future once these issues are fixed, and must reject.